# Sequence models for continuous cell cycle stage prediction from brightfield images

**Louis-Alexandre Leger**[1*] iD            LOUIS-ALEXANDRE.LEGER@EPFL.CH
**Maxine Leonardi**[1*] iD                MAXINE.LEONARDI@EPFL.CH
**Andrea Salati**[1*] iD                 ANDREA.SALATI@EPFL.CH
**Felix Naef**[1†] iD                   FELIX.NAEF@EPFL.CH
**Martin Weigert**[1,2†] iD           MARTIN.WEIGERT@TU-DRESDEN.DE

[1] *Institute of Bioengineering, School of Life Sciences, Ecole Polytechnique Fédérale de Lausanne (EPFL), Lausanne, Switzerland*

[2] *Center for Scalable Data Analytics and Artificial Intelligence (ScaDS.AI), Dresden/Leipzig, Germany*

**Editors:** Accepted for publication at MIDL 2025

## Abstract

Understanding cell cycle dynamics is crucial for studying biological processes such as growth, development and disease progression. While fluorescent protein reporters like the FUCCI system allow live monitoring of cell cycle phases, they require genetic engineering and occupy additional fluorescence channels, limiting broader applicability in complex experiments. In this study, we conduct a comprehensive evaluation of deep learning methods for predicting continuous FUCCI signals using non-fluorescence brightfield imaging, a widely available label-free modality. To that end, we generated a large dataset of 1.3 M images of dividing RPE1 cells with full cell cycle trajectories to quantitatively compare the predictive performance of distinct model categories including single time-frame models, causal state space models and bidirectional transformer models. We show that both causal and transformer-based models significantly outperform single- and fixed frame approaches, enabling the prediction of visually imperceptible transitions like G1/S within 1h resolution. Our findings underscore the importance of sequence models for accurate predictions of cell cycle dynamics and highlight their potential for label-free imaging.

**Keywords:** Cell cycle prediction, label-free microscopy, sequence-models

## 1. Introduction

The cell cycle is the driving force behind the growth and development of all living organisms. This well-studied sequence of cellular events is tightly regulated and aberrations in such mechanisms can lead to genomic instability, a key driver of various diseases including cancer (Kastan and Bartek, 2004). Live cell fluorescence microscopy has become a powerful tool for studying cell cycle progression, particularly through the genetic engineering of fluorescent reporters like the FUCCI system (Sakaue-Sawano et al., 2008; Stallaert et al., 2022). This system enables the distinction of cell cycle phases from single images by fluorescently tagging the two proteins Cdt1 and Geminin, whose expression changes distinctively with the cell cycle (Figure 1). Despite its utility, the classic FUCCI system and

---

*  Contributed equally

†  Shared supervision

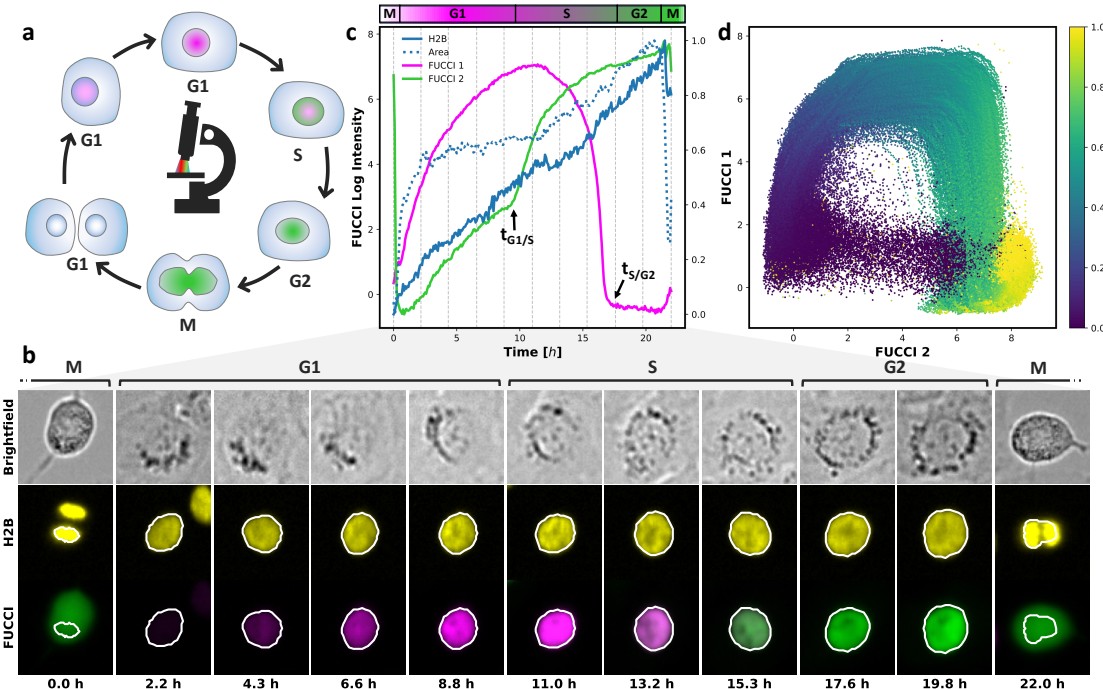

Figure 1: **Multi-modal imaging of Fucci-reporter cells reveals a continuous representation of cell cycle states. a)** Time-lapse imaging of Fucci-reporting cells allows for precise quantification of cell cycle staging through the characteristic oscillations of fluorescent reporter intensities. **b)** Representative time-lapse images of brightfield, H2B, and Fucci channels across one full (M-M) cell cycle. **c)** Quantification of integrated logarithmic fluorescence intensities from Fucci reporters, normalized nuclear area, and normalized total H2B signal in a representative full M-M (mitosis to mitosis) track. Vertical lines mark the time points corresponding to the images shown in d. **d)** Log-transformed Fucci manifold for continuous inference of cell cycle states.

recent variants (Sakaue-Sawano et al., 2017; Grant et al., 2018) are limiting in practice as they occupy two of the few available microscopy channels, reducing the ability to study other cellular processes simultaneously and requiring genetic modification that might interfere with the endogenous cell cycle regulation. In contrast, brightfield microscopy is an easily accessible and label-free imaging modality that does not require genetic engineering, providing limited specificity and imaging contrast. Although some cell cycle transitions, such as nuclear envelope breakdown, are marked by distinct morphological changes that are easily detectable, most cell cycle transitions are visually indiscernible in individual brightfield images of cells. In this paper, we ask whether leveraging the *temporal information* in time-lapse brightfield microscopy images of cells would allow to predict *continuous* cell cycle

states without the need for fluorescent reporters such as FUCCI. To address this, we study several sequence-based deep learning models, including transformers (Vaswani et al., 2017) and recently proposed state-space models (Gu et al., 2022). In particular, we will investigate both *causal* sequence models that only use information from previous time points, as well as *non-causal* models that may ingest the entire sequence. Providing a new dataset of over 1 M images of segmented and tracked RPE-1 cells with accompanying ground truth FUCCI signals, we show that both causal and transformer-based non-causal models significantly outperform single-frame approaches, enabling the prediction of morphologically subtle cell state transitions like G1/S within 1h resolution from live-cell brightfield imaging alone.

## 1.1. Related work

The prediction of individual cell states from single microscopy images has found considerable interest in the literature. For instance (Rappez et al., 2020) and (Narotamo et al., 2021; Li et al., 2024) showed that deep learning-based models allow to classify *discrete* cell cycle states such as the G1, S, or G2 phase from static multichannel images of cells. In particular, (Eulenberg et al., 2017; Blasi et al., 2016; Jin et al., 2021; He et al., 2022) explored the use of brightfield and phase-imaging data for cell cycle classification without fluorescence labeling. However, accurately annotating discrete cell cycle stages highly depends on the imaging modality with nuclear stains like DAPI or Hoechst providing clear features, while phase or brightfield imaging require substantial more manual annotation expertise. Beyond static images, several studies have investigated how the *temporal information* of images sequences can be leveraged to capture dynamic cell behaviors (Wang et al., 2020; Zhao et al., 2024; Chu et al., 2020). However, most approaches have so far been focused on well-defined cell phases such as mitosis, where large morphological changes facilitate annotation and prediction (Held et al., 2010; Moreno-Andrés et al., 2022; Jose et al., 2024; Su et al., 2017). A notable exception is (Ulicna et al., 2023) which applies dynamic time warping (DTW) to features from an unsupervised autoencoder, enabling to temporally align cell trajectories and to continuously predict cycle states across interphase. *Sequence models* such as recurrent neural networks and transformers (Vaswani et al., 2017) have been shown to be effective for modeling temporal data (Hewamalage et al., 2021; Wen et al., 2023), with *state-space models* such as Mamba (Gu and Dao, 2023) having recently gained considerable interest due to their training and inference efficiency. As the cell cycle is a continuous causal process, sequence models appear to be a natural fit for our task. However, studies that address this question systematically remain missing, with the closest work being (Jose et al., 2024) which uses recurrent neural networks for cell cycle prediction, but focuses on morphologically stark intermitotic phases.

## 2. Method

## 2.1. Dataset

We generated a large dataset of dividing human FUCCI RPE1 cells using joint brightfield and fluorescence time lapse microscopy. Movies spanning 72 hours were acquired at a 5 minute time resolution, capturing multiple cell cycles. In addition to the brightfield modality, we acquired a nuclear marker channel (Histone H2B) and the two FUCCI channels (FUCCI$_{1/2}$). Based on the H2B channel, we segmented the cell nuclei with a

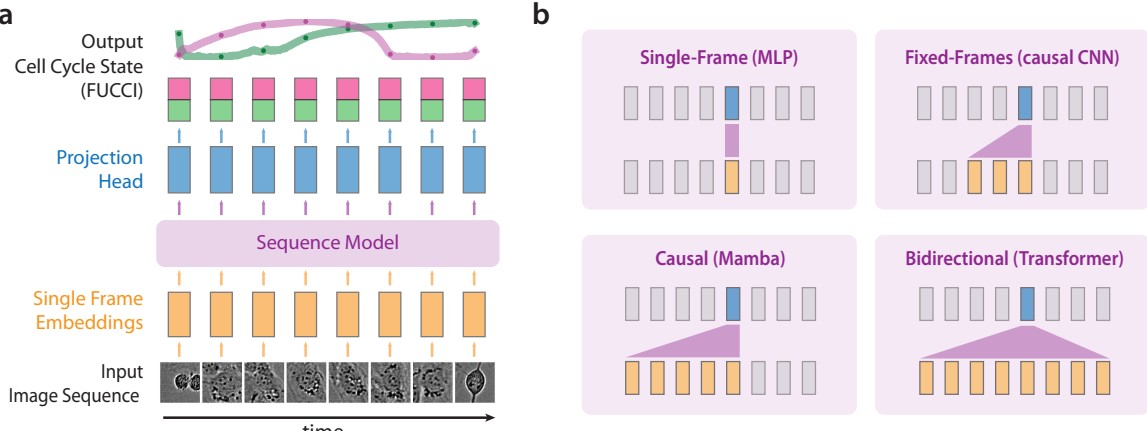

Figure 2: **Overview of approach. a)** We use a ResNet-18 (He et al., 2016) to extract single frame embeddings from an input sequences which are fed into a sequence model that predicts both Fucci channels. **b)** Sequence models explored in this paper: Single Frame MLP, Fixed-frame CNN, causal state-space models *e.g.* Mamba (Gu and Dao, 2023), bidirectional models *e.g.* transformers (Vaswani et al., 2017).

custom StarDist model (Schmidt et al., 2018) and tracked them across frames using Track-Mate (Tinevez et al., 2017). Note that since the amount of H2B histones needed by cells to pack DNA doubles during S (DNA replication), the H2B channel does contain information on cell cycle progression (Figure 1c). Below, we leverage this as a control to assess predictive performance of brightfield vs H2B. Full cell cycle tracks (from one mitosis to the next, M-M) were identified using K-Means clustering and ground-truth Fucci signals were computed by normalizing the average Fucci intensities measured across the segmented nuclear mask (Figure 1c,d). The training dataset comprises 5,188 full (M-M) cell cycle tracks with an average track length of 230 frames. Each track contains paired brightfield and H2B images of size $64 \times 64$ centered on the nucleus and the corresponding integrated Fucci signals. To evaluate model performance, we created two additional test datasets: Regular, which contains 358 additional full tracks from RPE1 cells acquired at similar conditions as the training set, and Drug, which comprises 73 complete tracks of cells treated with the cell cycle inhibitor *Palbociclib* that heavily distorts the cell cycle and which is used in the clinic to treat breast cancer. In total, this training and testing dataset consists of approximately 1.3 M images and Fucci signals, all of which we make publicly available alongside this paper[1].

## 2.2. Method

The cell cycle prediction task can be formalized as follows: Given a temporal sequence of $N$ brightfield images $X \in \mathbb{R}^{N \times 64 \times 64}$, each associated with a corresponding two-dimensional

---

1. https://zenodo.org/records/14987478

FUCCI signal $Y \in \mathbb{R}^{N \times 2}$, the goal is to train a model $f$ that predicts the normalized FUCCI intensities across the entire sequence in a supervised manner, *i.e.* $f : \mathbb{R}^{N \times 64 \times 64} \to \mathbb{R}^{N \times 2}$. Note that the length $N$ of the input is not fixed and the sequences is not required to span the entire cell cycle, allowing us to analyze the impact of temporal context on prediction performance. For this cell cycle prediction task, we evaluated three conceptually distinct model classes: single-frame models, causal models and non-causal models (Figure 2). Each model uses first a ResNet-18 (He et al., 2016) as feature extractor which for every input image in the sequence independently creates a 512-dimensional embedding. The sequence of embeddings is then fed into the proper sequence model head that differs between the three classes: *Single-frame models* serve as a baseline, predicting the FUCCI signal from each image embedding independently without leveraging temporal information. We use a simple 4 layer MLP with hidden dimension 512. *Fixed-frames models* that use a fixed history of past frames for prediction and for which we use a causal convolutional neural network (Van Den Oord et al., 2016) with a fixed causal temporal receptive field. *Causal models* that incorporate past and present temporal context in more flexible way and which are able to potentially capture arbitrary long temporal dependencies. In particular, we compare LSTMs (Hochreiter and Schmidhuber, 1997), Mamba (Gu and Dao, 2023), and causal transformers with masked attention. *Bidirectional models* process the entire sequence bidirectionally, using both past and future frames for inference (*i.e.* non-causal information). We use a standard transformer (4 layers) as representative architecture. To ensure a fair comparison, we choose all sequence heads to have the same number of parameters ($\approx 1M$). All transformer variants additionally use rotary positional embeddings (Su et al., 2024) to encode the relative temporal position of each frame. We train each model for 150 epochs while randomly sampling subtracks of variable lengths, using a learning rate of $10^{-4}$ and $L_1$ loss. We use standard data augmentation such as random rotations and flips, which is applied to the whole track.

### 2.2.1. PERFORMANCE METRICS

We evaluated model prediction by computing the mean $L_1$ error for each FUCCI channel across a given track. Further, we use the *dynamic time warping distance* $\Delta_{DTW}$ between the predicted and ground truth FUCCI signal that takes into account both the signal prediction error as well as the temporal misalignment between the two signals. We use the default $\Delta_{DTW}$ distance implementation from the `dtaidistance` package (Meert et al., 2020) with a penalty of 0.1. Additionally, we introduced two biologically meaningful cell cycle checkpoints (Figure 1) and measured the time difference between our predicted and observed checkpoints in minutes. The first checkpoint $t_{G1/S}$ is the onset of the Geminin (FUCCI 2) signal, marking the G1/S phase transition (Sakaue-Sawano et al., 2017) (additional information about how this checkpoint were calculated can be found in the appendix). While the classic FUCCI reporter does not provide an exact molecular landmark of the S/G2 transition, we use the disappearance of the Cdt1 (FUCCI 1) signal as an approximate landmark for evaluating S/G2 transition predictions $t_{S/G2}$. We use these two landmarks to categorize cells into discrete G1, S, and G2 phase classes for which we compute the F1-score of predictions. S phase is more constrained than G1, as genome duplication depends on DNA polymerase rates and replication origins. G1 is the key regulator of cell cycle duration and

Table 1: **Cell cycle prediction accuracy for brightfield on Regular.** Shown are mean and standard deviation of the $L_1$ error per Fucci channel and $\Delta_{DTW}$ for full tracks across Regular. $\Delta_{DTW}$ when using H2B images as input for comparison.

| Models | Brightfield | | | Histone H2B |
| --- | --- | --- | --- | --- |
| | $L_{1,FUCCI_1}$ | $L_{1,FUCCI_2}$ | $\Delta_{DTW}$ | $\Delta_{DTW}$ |
| Single Frame | $0.193 \pm 0.066$ | $0.146 \pm 0.045$ | $3.735 \pm 0.863$ | $2.595 \pm 1.201$ |
| Causal CNN | $0.157 \pm 0.078$ | $0.122 \pm 0.049$ | $2.468 \pm 0.917$ | $2.165 \pm 1.210$ |
| Mamba | $0.112 \pm 0.072$ | $0.091 \pm 0.049$ | $1.444 \pm 0.898$ | $1.426 \pm 0.949$ |
| Transformer | $\mathbf{0.066 \pm 0.038}$ | $\mathbf{0.062 \pm 0.037}$ | $\mathbf{1.285 \pm 0.553}$ | $\mathbf{1.155 \pm 0.612}$ |

is expected to be more variable. In Regular, the biological variability in phase duration ranges from a standard deviation of 1.6h for the S phase, to 2.2h for the G1 phase.

## 3. Results

### 3.1. Comparison of prediction accuracy across sequence models

We first compare the performance of the different sequence models on predicting the Fucci signals from brightfield images for full (M-M) cell cycle tracks. As seen in Figure 3a, the single frame model predicts extremely noisy signals that deviate substantially from the ground truth Fucci signal, whereas both causal and bidirectional models achieve qualitatively much better predictions on both Fucci channels and generally aligns with the expected trends

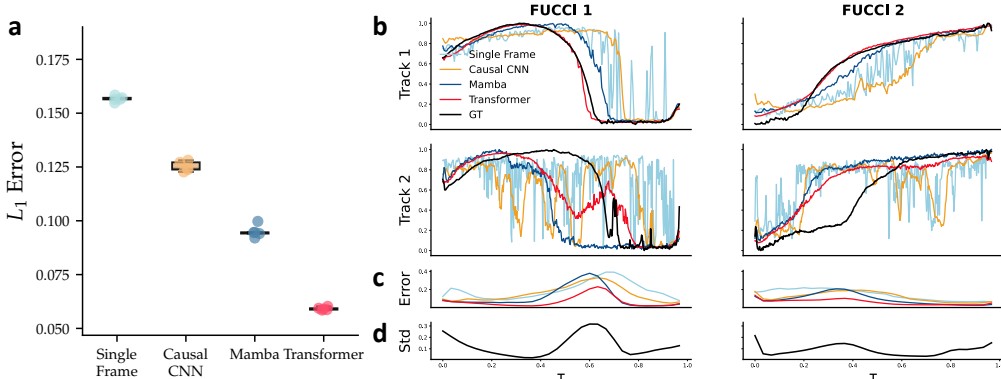

Figure 3: **Predictions on unperturbed RPE cells Regular. a)** Distribution of $L1$ errors across the different models. **b)** Predictions of Fucci signals on two example tracks: one with accurate and one with poor predictions. The ground truth signal is shown in black. **c)** Average prediction error, and **d)** variability of ground truth Fucci signals as a function of normalized cell cycle time $\tau$.

(*cf.* Supp. Figure 7a,b, Supp. Figure 8). To quantitatively assess the performance of the different models, we show the mean $L_1$ error for each FUCCI channel across all tracks as well as the average $\Delta_{DTW}$ distance between the predicted and ground truth FUCCI signals for the different models in Table 1. As expected, the single frame model which operates without integrating temporal information performs the worst across all metrics ($\Delta_{DTW} = 3.735$), while integrating the full bidirectional (non-causal) sequence information via a transformer achieves the best prediction ($\Delta_{DTW} = 1.285$). Surprisingly, there is a notable difference between the performance of the fixed-frames model (causal CNN) and the state-space model (Mamba), with the former performing substantially worse than the latter ($\Delta_{DTW} = 2.468$ vs. 1.444). This suggests that models that allow information propagation across the entire sequence can be more effective than models that only use a fixed-size temporal context. We additionally computed $\Delta_{DTW}$ when training with the H2B channel as input modality, which a priori should be a substantially easier task as the H2B signal is biologically correlated with the cell cycle. Indeed, this is corroborated by the performance of the single frame model that vastly improves in this case ($\Delta_{DTW} = 2.595$). Interestingly, both state-space models as well as the transformer only marginally improve, suggesting that these models are able to extract temporal cues from the brightfield images comparable to the easier H2B modality. The predictions at biological checkpoints remain consistent with the performance observed across other evaluation metrics (Table 2, Supp. Figure 9). As observed in Supp. Figure 6, the timing of these checkpoints is highly variable. Sequence models effectively capture this variability, with the bi-directional transformer predictions being more accurate and therefore better aligned with the ground-truth distribution. In contrast, other methods tend to underestimate such variability. As expected, the majority of prediction errors measured by the $L_1$ metric occur near the $t_{G1/S}$ and $t_{S/G2}$ landmarks, reflecting abrupt transitions where the FUCCI signal exhibits higher variability (Figure 3 b, c).

### 3.2. Prediction on partial tracks

So far we focused on full (M-M) tracks of non-perturbed cells, all of which exhibit fairly stereotypical cell cycle trajectories and for which sequence models are able to base their predictions on a well defined starting points (*i.e.* the cell division event). We now evaluate the performance of the different models on partial tracks, where the starting point is not known a priori, which is a more challenging task. In Figure 4, we show the average $L_1$ error with cropped partial tracks as input, indicated by their relative start and end time

Table 2: **Prediction accuracy of biological checkpoints and selected cell cycle states from brightfield images on REGULAR.**

| Models | $\Delta t_{G1/S}$[min] | $\Delta t_{S/G2}$[min] | $G1$ | $S$ | $G2$ |
|---|---|---|---|---|---|
| Single Frame | 191.3 | 117.9 | 0.71 | 0.64 | 0.87 |
| Causal CNN | 146.8 | 113.3 | 0.78 | 0.64 | 0.86 |
| Mamba | 111.4 | 102.4 | 0.83 | 0.72 | 0.89 |
| Transformer | **60.1** | **57.2** | **0.90** | **0.85** | **0.93** |

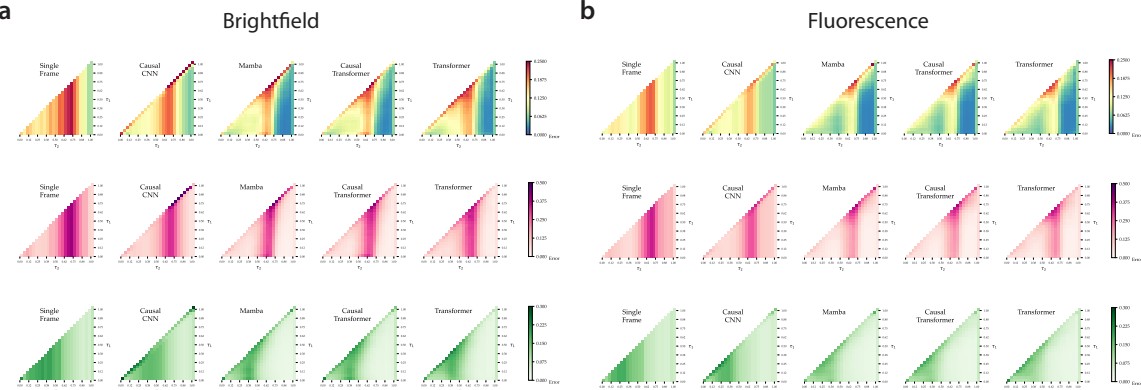

Figure 4: **Comparative performance on partial cell cycle tracks (brightfield).** Shown is the average $L_1$ error of both Fucci signals when using partial tracks as input, parametrized by their relative start and end time $\tau_1 \leq \tau_2 \in [0,1]$.

$\tau_1 \leq \tau_2 \in [0,1]$. These partial tracks ranged from single-frame portions (along the diagonal) to entire tracks (lower-right element). Causal models still achieved better accuracy in predicting Fucci values compared to non-temporal MLPs or fixed history CNNs. Surprisingly, the performance advantage of transformers over causal methods observed in full tracks diminishes on partial tracks. For all models, the error is maximal for segments that end near $t_{S/G2}$. Errors in Fucci$_1$ show the same pattern while the errors in the Fucci$_2$ arise mostly when taking segments from the beginning of the cell cycle (Supp. Figure 10).

### 3.3. Prediction on out-of-distribution perturbations

Finally, we evaluate the model performance on Drug, *i.e.* biologically strongly perturbed cells that can be considered out-of-distribution. Specifically, these cells were treated with the drug *Palbociclib*, a CDK4-6 inhibitor, that increases the cell cycle duration almost two-fold from ∼20h to 40 h (Figure 5 a) and specifically the G1 phase duration, leading to a strongly distorted cell-cycle (Figure 5 b). As expected, almost all models demonstrated a significant drop in accuracy when predicting Fucci signal in these unseen drug-treated cells, as indicated by all evaluation metrics (Table 3). The notable exception is the bidirectional transformer, that provides reasonable predictions and correctly captures distortions in the G1 phase (Figure 5 c) that all other models significantly underestimated. Interestingly, the MLP outperformed the other causal models on this distorted data, potentially as the latter overfitted on the training data. When performing the same analysis on H2B, we found slightly better predictive performance (Supp. Tables B 2 and B 3, Supp. Figure 11, Supp. Figure 12), which is expected due to the stronger correlation with the cell cycle.

### 4. Discussion and Conclusions

In this study, we generated and released a large dataset of cycling RPE1 cells under both normal and drug-treated conditions and used it to investigate the utility of sequence models to infer the continuous cell cycle state from label-free brightfield images. Our analysis

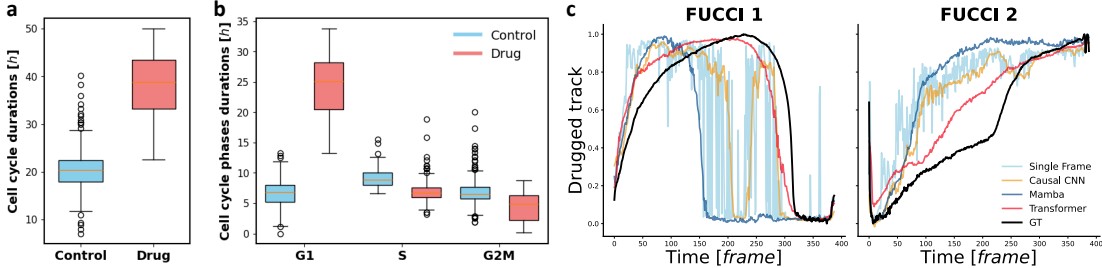

Figure 5: **Results on perturbed RPE cells DRUG. a,b)** Effect of CDK4/6 inhibition on cell cycle durations and onset of G1, S, and G2/M phases, extending G1 duration while leaving S and G2/M unchanged. **c)** Example FUCCI predictions on DRUG.

demonstrates that temporal sequence models can significantly improve the cell cycle prediction accuracy and enable the assessment of cell cycle state from brightfield images to a level comparable when using the more informative H2B channel. This suggests that brightfield morphological cues alone carry sufficient information for cell cycle inference. Importantly, we found that causal state-space models substantially outperform commonly used fixed-history convolutional networks, demonstrating their potential for real-time computer vision and smart microscopy applications, where such causal inference is essential (Simon et al., 2025; Mahecic et al., 2022). We note that the observed reduced accuracy on drug-treated cells underscores that creating general predictive cell state models for strong biological perturbations remains challenging. For broader applicability, the model needs exposure to more diverse cell types, microscopy setups, and biologically perturbed conditions. Capturing variations in morphology, imaging parameters, and drug-induced alterations will enhance robustness, allowing it to reliably detect atypical or rare cell cycle dynamics. Furthermore, it would be valuable to explore probabilistic models such as variational autoencoders (Zhao et al., 2019), that can provide uncertainty estimates of the predicted cell cycle state instead of point predictions as we currently do. Our findings demonstrate that sequence models can be effective predictors of cellular dynamics in more controlled settings, providing a powerful tool for studying proliferation, cancer dynamics, and addressing cell cycle-related confounding factors. With larger datasets, models could improve drug effect predictions, paving the way for screening and broader biomedical applications.

Table 3: $L_1$ **error and** $\Delta_{DTW}$ **for FUCCI channels on DRUG.**

| Models | $L_{1,FUCCI_1}$ | $L_{1,FUCCI_2}$ | $\Delta_{DTW}$ |
|---|---|---|---|
| Single Frame | $0.239 \pm 0.082$ | $0.182 \pm 0.056$ | $5.329 \pm 1.147$ |
| Causal CNN | $0.252 \pm 0.113$ | $0.161 \pm 0.059$ | $4.323 \pm 1.302$ |
| Mamba | $0.485 \pm 0.090$ | $0.259 \pm 0.045$ | $3.563 \pm 1.918$ |
| Transformer | $\mathbf{0.147 \pm 0.056}$ | $\mathbf{0.139 \pm 0.048}$ | $\mathbf{3.022 \pm 0.985}$ |

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

## Appendix A. Supplementary Methods

### A.1. Cell culture

FUCCI-RPE1 cells, kindly provided by Battich et al. [2020], were cultured at 37°C with 5% CO2 in DMEM/F12 medium (Gibco 11320033), supplemented with 1 % non-essential amino acids (NEAA) (Gibco 11140-035), 1% penicillin-streptomycin (Sigma-Aldrich G6784), and 10% fetal bovine serum (FBS) (Gibco 10437-028). In addition, the H2B-iRFP marker, driven by a PGK promoter, was introduced into the cells using the second-generation lentiviral system with a commercially available plasmid (Addgene: 90237).

### A.2. Imaging

For imaging, H2B-FUCCI-RPE1 cells were seeded into 96-well plates and cultured under the conditions described above, with Fluorobrite medium (Gibco A1896701) replacing DMEM/F12. For the perturbation experiments, cells were treated with 10 nM Palbociclib (CDK4-6 inhibitor). Images from four channels—Brightfield, H2B (far red), Cdt1 (red), and Geminin (green)—were acquired every 5 minutes using a PerkinElmer Operetta Microscope with a 20x/0.80 objective (wide-field microscopy). Four or nine tiles per well were captured for each channel, with a 15% overlap for subsequent stitching. In the images, 1 pixel equates to $0.5979761\mu m$. The laser intensities and time of exposure for each channel are shown in the table below.

| Channel | Laser intensity | Exposure time |
|---|---|---|
| FUCCI Green | 25% | 30ms |
| FUCCI Red | 15% | 10ms |
| H2B Far red | 30% | 30ms |
| Brightfield | 50% | 5ms |

### A.3. Image preprocessing

Image preprocessing involved stitching the tiles (Preibisch et al., 2009) and applying background subtraction to fluorescent channels using a rolling ball algorithm. Cell nuclei were segmented on the H2B channel with a custom StarDist model (Weigert and Schmidt, 2022) , and tracked across frames using TrackMate (Tinevez et al., 2017). Full cell cycle tracks (M-M, tracks encompassing one complete cell division cycle from one mitosis (M) to the next) were isolated using K-Means clustering of interpolated FUCCI signals. Our groundtruth labels were obtained by averaging the fluorescent channels over the nuclei area and taking the logarithm of this signal, with a smooth noise removal. The raw fluorescent FUCCI signal is not normalized for background noise (starting at $2^5$) and expresses a greater dynamic range in log scale as previously shown in DeepCycle (Rappez et al., 2020). However taking the logarithm increases the dynamic range of the background noise, leading to interesting questions about the proper scale of these tracks. We average the pixels present in the nucleus for each FUCCI marker and then for the background noise normalization, we express the signal shifted and in units of an $\epsilon$, where $\epsilon$ can be k-th percentile of the signals distribution. We use the 1st percentile $\epsilon = P_1 = (P_{1,f_1}, P_{1,f_2})$. Then to deal with the increased dynamic

range from the log, we take the Softplus with $\beta = 1$ of our new units of $\epsilon$ before applying the logarithm:

$$F = (f_1, f_2), \quad \overline{F} = \frac{F}{A} = (\overline{f_1}, \overline{f_2})$$

$$F' = \frac{\overline{F} - \epsilon}{\epsilon}, \quad \text{FUCCI} = log_2(Softplus(F'))$$

Where $F$ is the raw Fucci values from imaging of each nucleus pixel, $A$ is the area of the nucleus and $\text{Softplus}(x) = \frac{1}{\beta} * log(1 + exp(\beta * x))$.

### A.4. Identification of biological checkpoints

To identify the time frame at which phase transitions occur in a sequence of intensities (both in real data and in the model's predictions), we devised a simple threshold-based method that accurately detects the onset of the Fucci green signal (G1/S) and the disappearance of the red signal (S/G2).

- The "linear" signal (not log-transformed) of both channels is normalized between 0 and 1 to ensure comparability across tracks.

- The signal is smoothed using a convolution with a window size of 20.

- For the green signal, we identify the first time point where the intensity crosses above the 0.05 threshold (5% of its maximum intensity).

- For the red signal, we determine the transition point as the first time it drops below the 0.05 threshold (5% of its maximum intensity), marking its disappearance.

# Appendix B. Supplementary Tables

Table B 1: Number of parameters for each sequence model head.

| Model | Parameters |
|---|---|
| MLP | $1.32 \times 10^6$ |
| Causal CNN | $0.94 \times 10^6$ |
| LSTM | $1.28 \times 10^6$ |
| Mamba | $1.11 \times 10^6$ |
| Transformer | $1.12 \times 10^6$ |

Table B 2: **Side by side performance comparison of BF and H2B modalities at predicting Fucci channels on Regular.** Both data modalities present similar results: sequence encoders outperform the single frame method. Moreover H2B only shows modestly better performance than BF.

| Models | Brightfield | | | | Histone H2B | | | |
|---|---|---|---|---|---|---|---|---|
| | $L_{1,FUCCI_1}$ | $L_{1,FUCCI_2}$ | $R^2$ | $DTW$ | $L_{1,FUCCI_1}$ | $L_{1,FUCCI_2}$ | $R^2$ | $\Delta_{DTW}$ |
| Single Frame | $0.193 \pm 0.066$ | $0.146 \pm 0.045$ | $0.459 \pm 0.271$ | $3.735 \pm 0.863$ | $0.183 \pm 0.104$ | $0.130 \pm 0.064$ | $0.491 \pm 0.431$ | $2.595 \pm 1.201$ |
| Causal CNN | $0.157 \pm 0.078$ | $0.122 \pm 0.049$ | $0.608 \pm 0.294$ | $2.468 \pm 0.917$ | $0.154 \pm 0.105$ | $0.118 \pm 0.061$ | $0.586 \pm 0.415$ | $2.165 \pm 1.210$ |
| LSTM | $0.108 \pm 0.069$ | $0.087 \pm 0.047$ | $0.749 \pm 0.266$ | $1.527 \pm 0.814$ | $0.079 \pm 0.065$ | $0.075 \pm 0.044$ | $0.833 \pm 0.265$ | $1.467 \pm 1.161$ |
| Causal Transformer | $0.121 \pm 0.073$ | $0.094 \pm 0.048$ | $0.720 \pm 0.279$ | $1.728 \pm 0.811$ | $0.079 \pm 0.057$ | $0.079 \pm 0.042$ | $0.839 \pm 0.214$ | $1.552 \pm 0.955$ |
| Mamba | $0.112 \pm 0.072$ | $0.091 \pm 0.049$ | $0.739 \pm 0.282$ | $1.444 \pm 0.898$ | $0.074 \pm 0.056$ | $0.075 \pm 0.040$ | $0.853 \pm 0.215$ | $1.426 \pm 0.949$ |
| Transformer | $\mathbf{0.066 \pm 0.038}$ | $\mathbf{0.062 \pm 0.037}$ | $\mathbf{0.892 \pm 0.111}$ | $\mathbf{1.285 \pm 0.553}$ | $\mathbf{0.056 \pm 0.039}$ | $\mathbf{0.054 \pm 0.033}$ | $\mathbf{0.912 \pm 0.116}$ | $\mathbf{1.155 \pm 0.612}$ |

Table B 3: **Performance metrics for both brightfield and histone H2B modalities on Drug.**

| Palbociclib Models | Brightfield | | | | Histone H2B | | | |
|---|---|---|---|---|---|---|---|---|
| | $L_{1,FUCCI_1}$ | $L_{1,FUCCI_2}$ | $R^2$ | $\Delta_{DTW}$ | $L_{1,FUCCI_1}$ | $L_{1,FUCCI_2}$ | $R^2$ | $\Delta_{DTW}$ |
| Single Frame | $0.239 \pm 0.082$ | $0.182 \pm 0.056$ | $-0.297 \pm 1.064$ | $5.329 \pm 1.147$ | $0.183 \pm 0.050$ | $0.107 \pm 0.048$ | $0.260 \pm 0.466$ | $3.285 \pm 0.820$ |
| Causal CNN | $0.252 \pm 0.113$ | $0.161 \pm 0.059$ | $-0.353 \pm 1.459$ | $4.323 \pm 1.302$ | $0.149 \pm 0.042$ | $0.125 \pm 0.040$ | $0.401 \pm 0.376$ | $3.077 \pm 0.840$ |
| LSTM | $0.424 \pm 0.101$ | $0.229 \pm 0.045$ | $-1.663 \pm 1.727$ | $3.685 \pm 1.678$ | $0.140 \pm 0.051$ | $0.115 \pm 0.040$ | $0.503 \pm 0.404$ | $\mathbf{2.750 \pm 0.838}$ |
| Causal Transformer | $0.326 \pm 0.104$ | $0.214 \pm 0.049$ | $-0.728 \pm 1.558$ | $5.159 \pm 1.562$ | $0.132 \pm 0.034$ | $0.111 \pm 0.037$ | $0.628 \pm 0.207$ | $3.154 \pm 0.872$ |
| Mamba | $0.485 \pm 0.090$ | $0.259 \pm 0.045$ | $-2.244 \pm 1.949$ | $3.563 \pm 1.918$ | $0.185 \pm 0.078$ | $0.134 \pm 0.043$ | $0.255 \pm 0.651$ | $2.896 \pm 0.903$ |
| Transformer | $\mathbf{0.147 \pm 0.056}$ | $\mathbf{0.139 \pm 0.048}$ | $\mathbf{0.408 \pm 0.478}$ | $\mathbf{3.022 \pm 0.985}$ | $\mathbf{0.074 \pm 0.029}$ | $\mathbf{0.095 \pm 0.031}$ | $\mathbf{0.789 \pm 0.131}$ | $2.896 \pm 1.201$ |

## Appendix C. Supplementary Figures

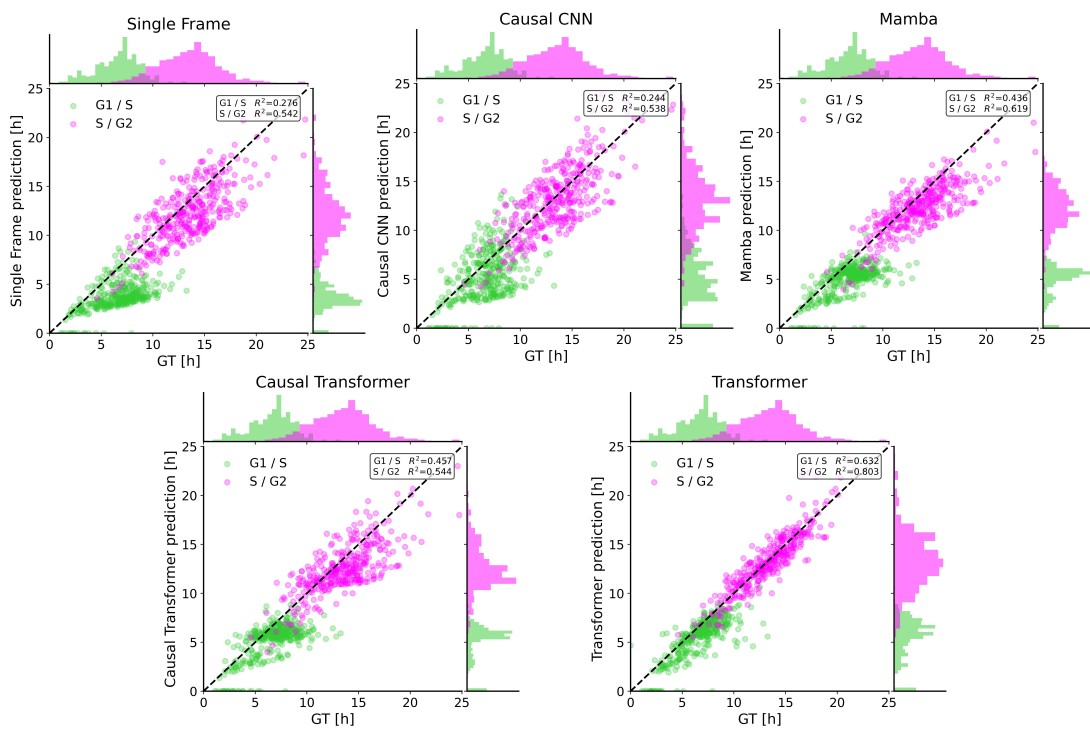

Figure 6: **Distribution of Phase Transition Timings in Ground Truth and Model Predictions Across Different Temporal encoders for Brightfield Imaging** The joint distribution of GT and predicted timings is here represented for the REGULAR data. The non-causal transformer is able to outperform the other temporal encoders.

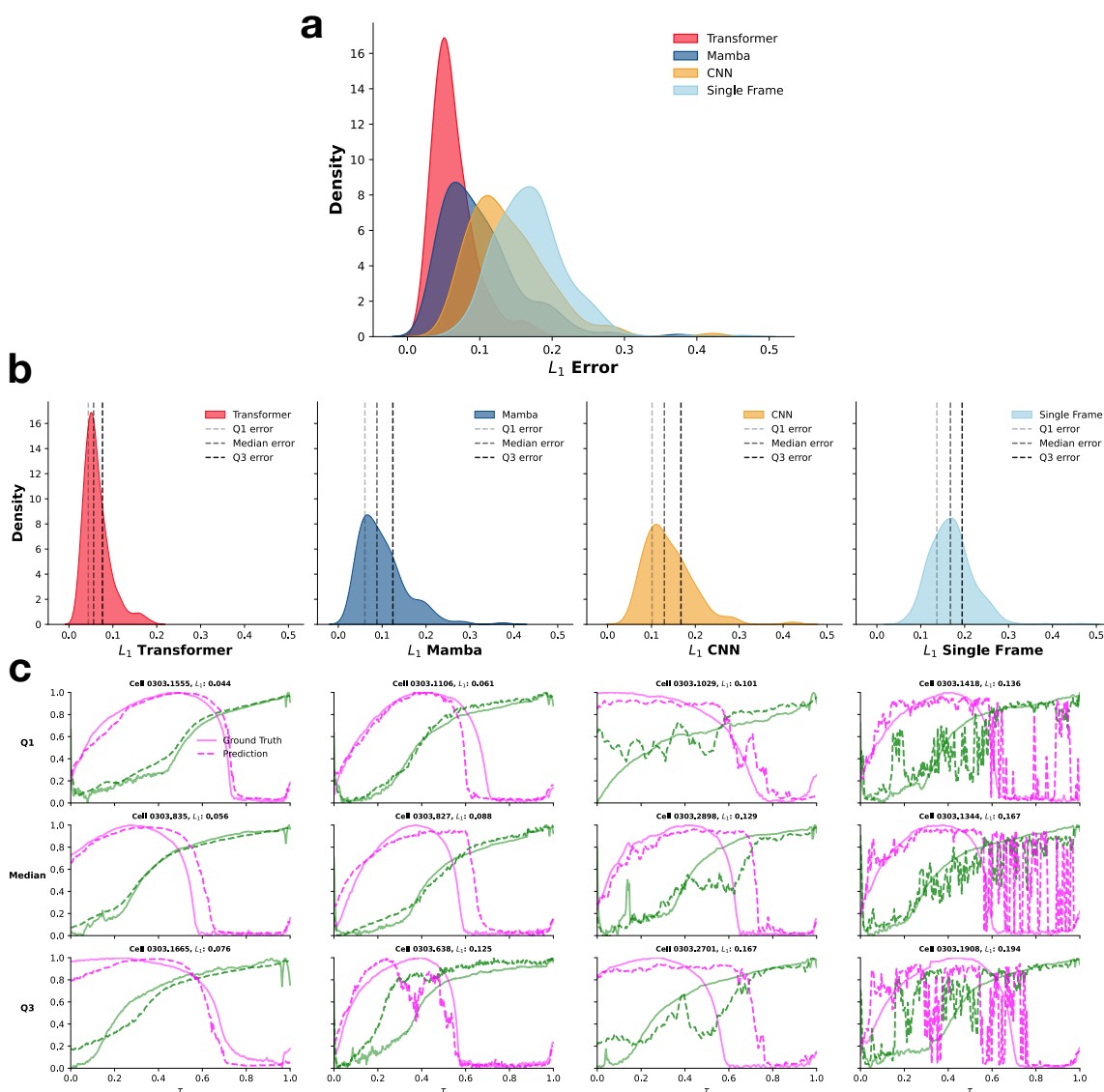

Figure 7: **Error Distribution of Predictions of FUCCI on Test Set with Brightfield.**
**a.** Distribution of L1 errors across the different models **b.** Error Distributions
with Q1, Median and Q3 Percentiles overlayed **c.** Q1, Median and Q3 Error
Predictions visualized per model.

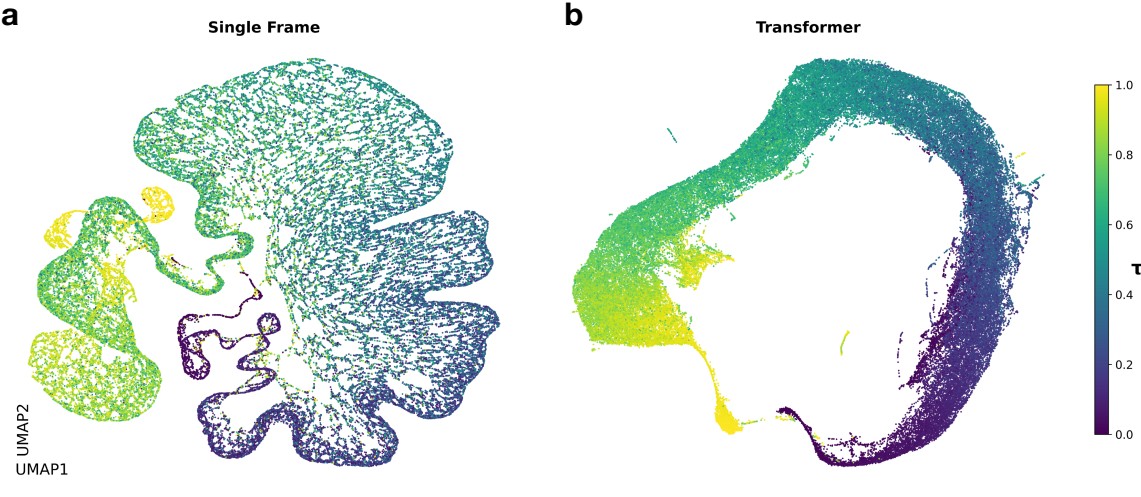

Figure 8: **Learned Latent Space Representations (UMAP).** Each frame of a track is represented as a dot in umap space, the coloring is the normalized time **a.** Single Frame (no history). **b.** Transformer (full sequence).

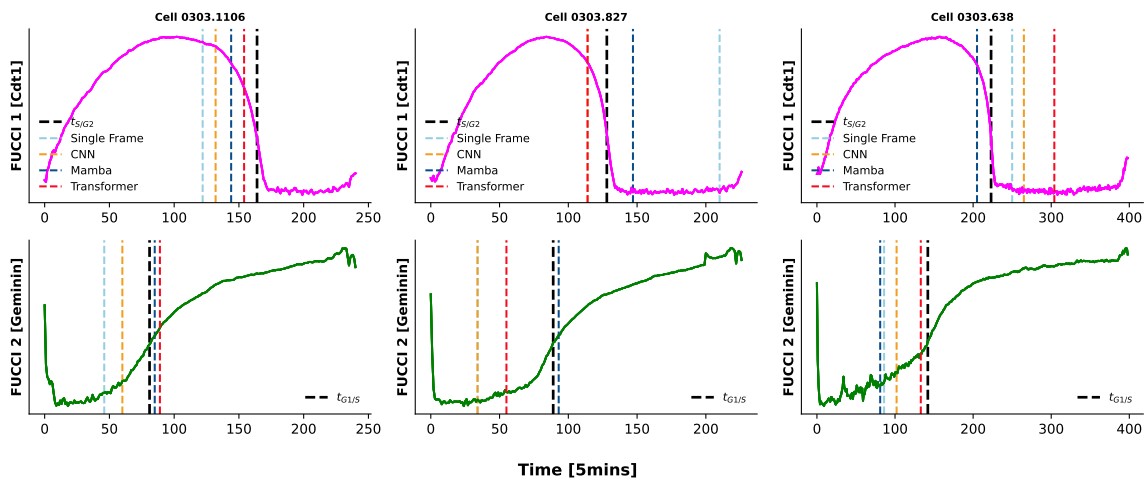

Figure 9: **Predicted $\Delta t_{G1/S}$ and $\Delta t_{S/G2}$ from BF images for the different models.**

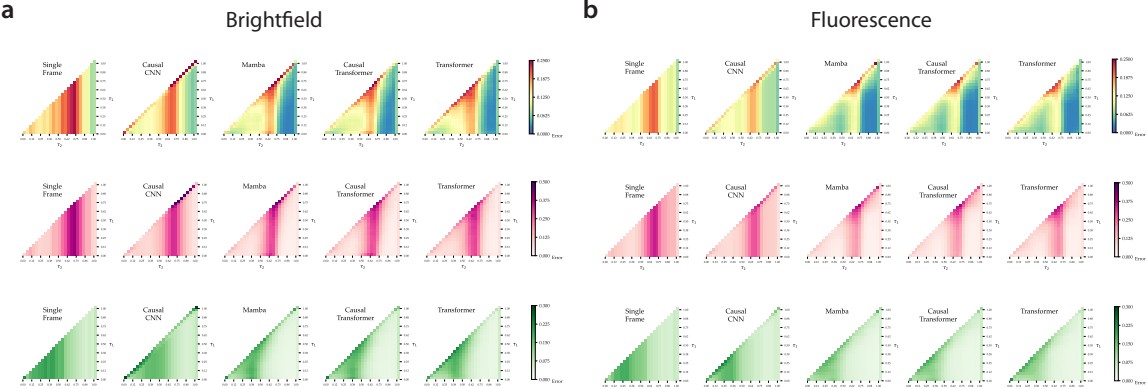

Figure 10: **Comparative Performance of Temporal Encoders in Predicting Fucci 1 and Fucci 2 from BF and H2B in partial cell cycle tracks on Regular.** Error maps showing the prediction error of the different models, assessed on the last frame of segments from the M-M track, spanning indices $\tau_1$ to $\tau_2$.

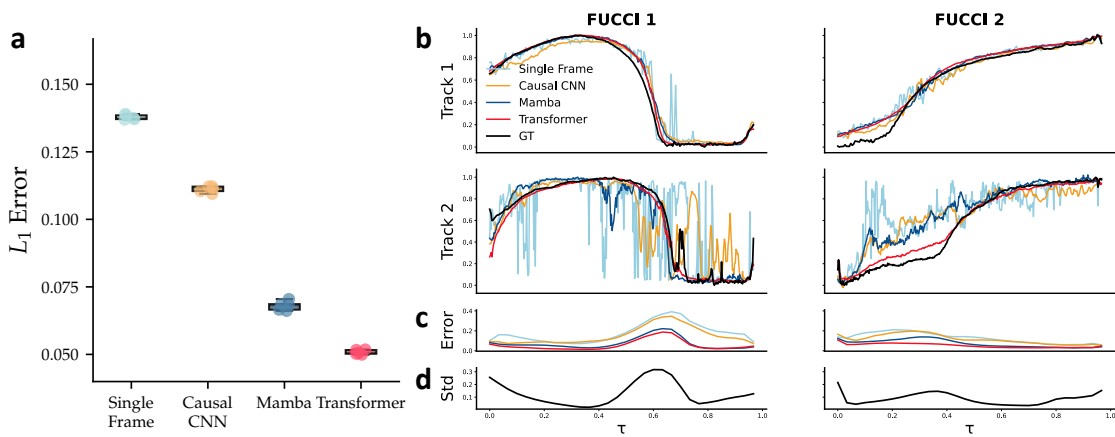

Figure 11: **Comparative Performance of Temporal Encoders in Predicting Continuous Cell Cycle States from H2B Imaging in Unperturbed RPE Cells. a)** Distribution of L1 errors across the different models. **b)** Example predictions of Fucci signals from different models on two tracks: one with accurate predictions and one with poor predictions. The ground truth signal is shown in black. **c)** Average prediction error and **d.** ground truth standard deviation are plotted in function of cell cycle phases.

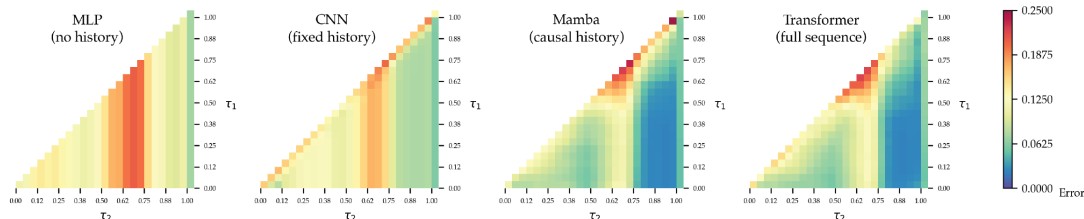

Figure 12: **Comparative Performance of Temporal Encoders in Predicting Continuous Cell Cycle States from H2B in partial cell cycle tracks.** Error maps showing the prediction error of the different models, assessed on the last frame of segments from the M-M track, spanning indices $\tau_1$ to $\tau_2$.

