# OpenReview forum: "Sequence models for continuous cell cycle stage prediction from brightfield images"
_MIDL.io/2025/Conference — MIDL 2025 Oral_

### Official Review · Reviewer_482n · 2025-02-21

**Confidence:** 5
**Preliminary Rating:** 5
**Recommendation:** Best Paper Award, Oral
**Final Rating:** 5

**Summary:**

Leger et al. address the task of predicting the cell cycle stages for a single cell in a continuous manner from unlabelled brightfield microscopy images. For this, they propose a series of deep-learning methods that go from the simplest approach of single-frame prediction to sequential-type methods integrating temporal information into the learning and prediction process. With this work, the authors show that (1) the cell cycle dynamic process and its discrete stages are predictable without fluorescence specificity, and (2) that performing this task accurately is possible using sequence models.

**Strengths:**

- Data-driven methodologies that integrate temporal information, such as transformers, are underdeveloped and yet not easily accessible in the field. Works like the one presented here are very much needed to enable their use for biological discovery.
- The authors elegantly prove their method by setting a series of logical comparisons:
  - The need for sequential approaches to predict the continuous cell cycle staging process from brightfield images is proven by comparing them to the "equivalent" single-frame approach (same image encoder) and setting an improvement baseline with the results obtained from using a fluorescent nuclear marker instead of brightfield images.
  - The capacity to infer the cell cycle stage of the proposed methods is further assessed by analyzing partial tracks and removing the strong prior of all the tracks starting at the mitotic event (M stage).
  - The proposed methods are challenged to infer the cell cycle stages on perturbed cell phenotypes, unseen by the models during training.
- The causal and bidirectional approaches, which can have different applicabilities, seem to perform accurately, and similarly with partial tracks.
- The methods are assessed considering the canonical L1 and accuracy metrics, but also biologically relevant aspects such as the stage transition.
- The authors have made the training data available, which is important for further developments and benchmarks of this type of work.
- The manuscript is well written, and all the results and methods are presented clearly.

**Weaknesses:**

I find no remarkable weakness in the presented problem nor in the methods proposed or the results obtained by the authors. Please, see my comments below that I encourage the authors to address. Most of them are minor issues or clarifications in the text.

**Detailed Comments:**

- Page 1: Would it be "This well-studied sequence of cellular events is tightly regulated, **AND** aberrations in
such mechanisms can lead to genomic instability,..."?
- Page 1 (the comma): "...two proteins Cdt1 and Geminin, whose expression changes distinctively..."
- Page 3: " enabling the prediction of morphologically important cell state transitions". Do you mean physiologically or biologically important? If not, could you elaborate on why these transitions are morphologically important and how you relate the model's prediction with cell morphology?
- Page 3: "..., yet for the case of a small number of discrete cell states". What cell states are you referring to? Are they M, G1, S and G2?
- Page 4: "... and non-causal models (cf. Figure 2)." Is the "cf" a typo?
- Page 5: "... we use a convolutional neural network **(s)**...". Remove the "s".
- Reference format seems inconsistent, being some without the year specified along the text.
- How are the G1/S and S/G2 transitions calculated or annotated? I would include this explicitly in the text for replicability.
- In the data augmentation of sequential models, are all the frames from a track altered with the same transformation or is it frame by frame?
- What type of fluorescence microscopy is applied? Is it widefield? Also, what are the pixel sizes? Could you include some metadata at least in zenodo?
- Figure 1: Authors claim that H2B is related to the cell cycle stages. I would agree that it may be related to M and the increase in the nuclear size, yet this is not directly relatable to all the stage transitions. It would help to guide the readers if the H2B signal is also displayed in c). Fucci --> FUCCI in the caption.
- Figure 3 would improve by indicating where the G1/S and S/G2 transitions occur and displaying the cells of Track 1 and 2. This would allow seeing if their shape is representative of more challenging cases to process. Additionally, it would be interesting to see where these two cells are located in the UMAPs from Figure 7. The same applies to the example used in Figure 5.
- How were the lengths of the tracks of perturbed cells normalized to fit in the sequential models? is there any specific transformation for the bidirectional or the causal models?

**Justification Of The Final Rating:**

I thank the authors for addressing all the comments in this review and updating the manuscript accordingly. I also take the opportunity to congratulate them for their work.

Best,
Estibaliz Gómez de Mariscal

**Justification Of The Preliminary Rating:**

Overall, the manuscript is of high quality and my highlights are related to clarity in the text and little Figure improvements. The justification for my rating is given by the strengths of the manuscript and significance cited above.

**Questions To Address In The Rebuttal:**

All the highlights in the detailed comments.

I have a major concern regarding the results with perturbed RPE cells: Under normal physiological conditions, the cell cycle is a well-characterized process in time. It would not be surprising that the sequential models learned these timings, and for example, would predict stage S after 6h with high confidence, independently of the cell's appearance. When looking at Figure 5's results, it feels that this may be happening with the causal model, which would imply that there are not (or it may not have learned) dynamic morphological features encoding the cell cycle stages. On the other hand, when looking at the transformer, it seems that it can accurately predict it. Could the authors double-check that this is not due to any normalization of the tracks before being analyzed with the network? Related to this, would it make more sense to assess the performance of the models with the partial tracks of perturbed cells?

More as a curiosity, the type of input images are crops around the cell nucleus and often miss part of the cytoplasm. Also, the type of cells chosen are quite flat and roundish compared to some highly motile cell phenotypes and the cell nucleus staining prediction (similar to H2B markers and related to the cell cycle) is possible from brightfield images. Could then the authors comment on whether the proposed cell cycle prediction is limited to this type of cell, conditions, and resolution?

**Special Issue:**

Yes

---

> ### Author Response · Authors · 2025-03-08
>
> We thank the reviewer for their positive comments and the constructive feedback!
>
> **Typos and grammatical mistakes**
>
> We really appreciate the detailed reporting of typos/mistakes and reference inconsistencies, all of which we have now corrected in the main text.
>
> **Details on G1/S and S/G2 transitions calculations**
>
> We thank the reviewer for pointing this out. We now added a section of the supplementary text where we explain it. It is a simple threshold based method that was empirically derived.
>
> **Data augmentation of sequential models, same or frame-by-frame?**
>
> Thanks for pointing this out. Yes, the same augmentation is applied to the whole track, this is now clarified in the manuscript.
>
> **Type of microscope and metadata**
>
> We used a widefield microscopy with pixel size of 0.598 um. We now have provided additional microscopy metadata in the supplementary text and on Zenodo:
>
> **Figure 1: Relation of H2B and cell cycle stages**
>
> This is a good suggestion. Indeed the reviewer is correct: the total nuclear H2B signal increases with cell cycle progression, similarly as the nuclear size but the two signals are not redundant (see new Fig. 1c), consistent with the literature (Fidorra, 1981; DOI: 10.1007/BF01324188). As suggested, we have now added the H2B signal and nuclear area as a function of time in the updated Figure 1c.
>
> **Normalization of track length for sequence models**
>
> We did not normalize the length of the tracks to a fixed length and all models ingest the full sequence of images (of different length) for each track to produce the FUCCI prediction of the same length as the input. As such, we don't need to perform rescaling or warping of the track sequences.
>
> **Performance on perturbed RPE cells and track normalization (predicting Stage S after 6h independent of cell appearance)**
>
> This is  an interesting point. The predictions from the sequential models do reflect a certain amount of memorization, depending on the model. However, the durations of G1 and other phases exhibit some variability in unperturbed cells, which we now highlight in a companion figure to Table 1 (Supp. Fig. 6). In this figure, we show how the measured and predicted G1/S​ and S/G2 times correlate in the different models. The strong positive correlations indicate that the models, including the causal ones, do not simply predict transitions at fixed times but instead capture this biological variability. That said, we do observe a bigger tendency for causal models to infer certain transitions at specific times (ex: G1/S at 6h), an effect that is largely mitigated by the transformer, that clearly shows the best correlation performance.
>
>
> We now address this more precisely in the manuscript:
>
> *As observed in Supp. Figure 6  the timing of these checkpoints is highly variable. Sequence models effectively capture this variability, with transformer predictions being more accurate and therefore better aligned with the ground-truth distribution. In contrast, other methods tend to underestimate such variability.*
>
>  As for normalization, each frame is independently normalized to have zero mean and unit standard deviation, so this should not introduce any issues. We also have the limitation that we currently only have a limited number of complete drug-tracks.
>
>
> **Small cell crops and limitation to cell type/condition**
>
> We indeed used a relatively small 64×64 pixel crop around the nucleus, which limits access to cytoplasmic morphological information. We chose to use this small field of view primarily to avoid introducing signal from neighboring cells in the typically crowded scenario.
>
> The reviewer also raised an important point regarding our analysis being limited to a single cell type. Generalizing to other cell types is a key question we aim to address in future work. We have already begun to generate datasets with other cell types with distinct morphology (fibroblasts, ES cells,...) that we plan to include in a follow-up study. Increasing the diversity of our training data would enhance both the accuracy and robustness of our model. We have now clarified this in the manuscript discussion. For broader applicability, the model needs exposure to more diverse cell types, microscopy setups, and biologically perturbed conditions. Capturing variations in morphology, imaging parameters, and drug-induced alterations should enhance robustness, allowing it to reliably detect atypical or rare cell cycle dynamics.

---

### Official Review · Reviewer_T5Cg · 2025-02-22

**Confidence:** 4
**Preliminary Rating:** 4
**Recommendation:** Oral, Poster

**Summary:**

The paper introduces a new framework for inferring continuous cell cycle stages solely from label-free brightfield time-lapse images, using fluorescent Fucci signals as the ground truth. Specifically, the authors construct a large-scale dataset (over 1.3 million images) of RPE1 cells with nuclear (H2B) and Fucci signals, then systematically compare different deep sequence models: single-frame MLP, fixed temporal convolution (Causal CNN), causal state-space (Mamba, LSTM), and bidirectional transformer. They report that bidirectional and causal sequence models significantly outperform single-frame baselines, capturing subtle cell cycle transitions, such as G1/S, with around one-hour resolution. A key strength is demonstrating that morphological cues in brightfield can rival more explicit nuclear markers (like H2B) when equipped with powerful sequence encoders. The results highlight important implications for label-free monitoring of cell cycle states and open avenues for broader “smart” microscopy applications where fluorescent labeling is impractical or limited.

**Strengths:**

1. The authors provide a large dataset of labeled, time-lapse brightfield and fluorescent images (1.3 M images total) featuring multiple full cell cycles.
2. They systematically compare a wide range of sequence models—from single-frame MLP to state-of-the-art transformers—using consistent architectural sizes and metrics (L1 error, DTW distance, checkpoint times).
3. Predicting cell cycle phase transitions from standard brightfield images can be highly useful in labs lacking advanced or multiple fluorescence channels.
4. The paper includes rigorous metrics (including biologically meaningful G1/S and S/G2 transitions) and also tests on out-of-distribution data (CDK4-6 inhibitor). These analyses strengthen the conclusions about sequence model robustness and generalizability.
5. The manuscript is well-structured, with carefully presented figures, sufficient detail on model architectures, and a thoughtful discussion of biological checkpoints.

**Weaknesses:**

1. While the transformer and Mamba approaches are strong, the paper could benefit from a brief discussion of possible enhancements like hybrid multi-scale convolution-transformer backbones or variational latent models for capturing subtle morphological transitions.
2. The paper emphasizes predictive performance but offers relatively little about how the sequence models learn to capture morphological signals of the cell cycle.

**Detailed Comments:**

1. The authors use L1 loss with a time-warped DTW metric for validation. Clarifying how hyperparameters in DTW (e.g., warping penalties) were selected would be beneficial.
2. The G1/S and S/G2 transitions are inferred from Fucci intensities. Including a short discussion on the biological variability in these landmarks may help interpret the model’s timing offsets.
3. Sequence models like Transformers can be GPU-intensive for long time series. The authors might note memory usage or speed trade-offs, guiding future large-scale experiments.
4. Since the dataset is publicly available, releasing checkpoints of the best performing models (or code) would accelerate community benchmarking.
5. The model’s improved generalization for the transformer is promising. More insights into which morphological features shift under Palbociclib would be a valuable extension.

**Justification Of The Preliminary Rating:**

This paper offers a valuable dataset and a thorough comparison of sequence models for continuous cell cycle prediction from brightfield images, a task with strong relevance in bioimaging and drug discovery. The key novelty lies in the temporal modeling of Fucci-like signals without needing actual fluorescent labeling, and the authors demonstrate significant gains over single-frame baselines. While certain aspects—such as testing on more cell lines, deeper interpretability, and partial-track complexities—could be expanded, the overall methodology is great, and results are compelling.

**Questions To Address In The Rebuttal:**

1. Could the authors provide more analysis of which morphological features the Transformer or Mamba exploit, for instance via attention maps or saliency approaches?
2. How feasible is it to run the best-performing transformer in real-time or near real-time? Are there any latency concerns for live-cell experiments?
3. What do the authors propose to mitigate the diminished improvements in partial-tracking scenarios? Could a hierarchical approach or prior knowledge about morphological transitions help?

---

> ### Author Response · Authors · 2025-03-08
>
> We thank the reviewer for their detailed comments and valuable suggestions!
>
> **The paper could benefit from a brief discussion of [model] enhancements**
>
> This is a good point. Indeed, incorporating latent or probabilistic models capable of predicting valid posteriors (or sampling from them) for cell cycle phases -- rather than making simple point predictions, as we currently do -- would be highly valuable. Variational autoencoders for regression could be a promising approach in this direction. We have added a discussion of this in the manuscript.
>
> **The paper [...] offers relatively little about how the sequence models learn to capture morphological signals**
>
> We agree that examining the explainability of model predictions related to morphological features would be very interesting and potentially linking these features to the cellular context would make it truly insightful and interpretable biologically. However a proper in-depth investigation of this would require significantly more effort, as robust and generalizable explainability methods particular for sequential image data as ours are still lacking in our field. This work primarily serves to demonstrate that cell cycle inference from brightfield image sequences is feasible. A detailed feature attribution analysis falls beyond its scope, but we plan to address broader biological questions and extend the analysis to multiple cell types in future work.
>
>
> **1. Clarifying hyperparameters in DTW**
>
> We used the `dtadistance` python library (`dtaidistance.dtw_ndim.distance_fast`) with all standard parameters except that the penalty was set to 0.1, which we chose empirically by qualitatively evaluating the alignment on a few individual cell tracks.
>
> **2. Biological variability of G1/S and S/G2 transitions**
>
> In general we expect the duration of S to be more constrained than G1, since it takes a certain minimum of time to duplicate an entire mammalian genome. In contrast, G1 duration serves as the primary regulatory mechanism for controlling cell cycle length, which we do see in our data. We added a discussion of this to the methods section:
>
> *S phase is more constrained than G1, as genome duplication depends on DNA polymerase rates and replication origins. G1 is the key regulator of cell cycle duration and is expected to be more variable. In \dataregular, the biological variability in phase duration ranges from a standard deviation of 1.6 hours for the S phase, to 2.2 hours for the G1 phase.*
>
> To further clarify this, we added a new supp section on how the two checkpoints/landmarks were calculated (A.4.) and we included a new figure (Supp. Fig. 6) depicting the correlations between the measured and predicted transition times.
>
> *As observed in Supp. Figure 6  the timing of these checkpoints is highly variable. Sequence models effectively capture this variability, with transformer predictions being more accurate and therefore better aligned with the ground-truth distribution. In contrast, other methods tend to underestimate such variability.*
>
>
> **3. memory and speed-trade off for transformers**
>
> All sequence models in our approach operate on sequences of single-frame embeddings that are extracted from the input images by a ResNet-18. Since this results in relatively short and small input sequences (512 features per frame, 200-400 frames), both inference speed and memory consumption remain low (~15ms per track, ~500MB of GPU memory) for all models.
>
> **4. releasing checkpoints of the models**
>
> We agree and updated our Zenodo repository to include the model checkpoints.
>
> **5. Morphological features shift under Palbociclib**
>
> We agree that this would be very interesting to explore and is very related to the point raised before by the reviewer regarding explainability. We qualitatively did not observe noticeable differences in the morphological appearance of single images between the REGULAR and DRUG datasets, but we agree that this merits further investigation.
>
> **More analysis of which morphological features the Transformer or Mamba exploit**
>
> See our comments above.
>
> **How feasible is it to run the best-performing transformer in real-time?**
>
> As the current approach relies on the proper detection and tracking cells, this relies on the availability of fast and robust segmentation and tracking methods and on the number of cells in the field of view. As the typical temporal sampling for our datasets is ~5 minutes, this should be feasible.
>
> **Partial-tracking: Could a hierarchical approach or prior knowledge about morphological transitions help?**
>
> We have not explored this in detail, but it is an interesting idea. The improved performance of the non-causal transformer on partial tracks indeed suggests that bidirectional integration of information from certain checkpoints is beneficial, which might be leveraged in a more hierarchical (and biologically appropriate) representation of the latent cell phase within any model.

---

> ### Comment · Area_Chair_eC4i · 2025-03-19
> **Final review**
>
> Dear reviewer, can you please update your review to include a final rating after reading the rebuttal/other reviews? Thanks!

---

### Official Review · Reviewer_LwDu · 2025-02-23

**Confidence:** 4
**Preliminary Rating:** 5
**Recommendation:** Oral
**Final Rating:** 5

**Summary:**

The article investigates whether deep learning models can predict cell
cycle stages from brightfield microscopy images without requiring
fluorescent reporters, which require genetic modifications and
separate channels for imaging. To this end, single image-based
prediction is compared with state-space (Mamba) and transformer models
that take into account a sequence of images for prediction. Authors
further assess the difference between temporally causal models and
non-causal models.

**Strengths:**

+ The question is very interesting and the approach for the empirical
  analysis well designed.
+ The test set is chosen very carefully. Authors first use a "regular"
  set of cells that go through the normal cell cycle. This is a good
  test set but it does not rule out the possibility that the model is
  simply memorizing the output through biases. The second set, which
  includes cells with disturbed cycles is a perfect test
  set. Empirical tests here rule out the possibility that the model
  memorizes the output through its biases.
+ The dataset is also released as part of this work. This can
  accelerate research in this direction.
+ Results on the "regular" dataset are very good. It looks like
  transformer-based models can actually do such prediction.
+ Difference between Mamba and Transformer models is noteworthy.

**Weaknesses:**

+ Results on the "drug" dataset is not very high. This may indicate
  that the models do memorization to some extent. However, it is also
  clear that this is not pure memorization. That said, the conclusions
  and discussions well describe the situation.

**Detailed Comments:**

I think this is a solid article. I recommend authors perhaps to add
some discussions on the applications these models can be used. For
instance, given the degraded performance on the "drug" dataset means
that deep learning models are not yet ready to be used for studying
effects of drugs or in general cells with disturbed
cycles. Highlighting the application areas they can be used in would
be useful.

**Justification Of The Final Rating:**

Excellent work. Authors rebuttal only strengthened my opinion about this article.
I think this is a good contribution to the conference and I support an oral presentation for this article.
I should also thank the authors.

**Justification Of The Preliminary Rating:**

This is a solid empirical analysis article. Authors do a great job in
evaluating the models as well as highlighting the limitations. The
article comes also with a dataset, which is plus. I think the article
overall will lead to a great discussion in the conference.

**Questions To Address In The Rebuttal:**

Please see above.

**Special Issue:**

No

---

> ### Author Response · Authors · 2025-03-08
>
> We thank the reviewer for their constructive feedback and positive evaluation. Below are our responses to the detailed points raised.
>
> **Results on the "drug" dataset are not very high indicating memorization**
>
> Our results on the DRUG dataset indeed suggest that some models, particularly Mamba, exhibit a certain degree of memorization. However, the duration of G1 and other phases varies even in unperturbed cells, which we now highlight in a new companion figure (Supp. Fig. 6) to Table 1. This figure illustrates how the measured and predicted G1/S and S/G2 transition times correlate. These correlations indicate that the models, including causal ones, do not merely predict transitions at fixed times (i.e., sequence positions) but instead capture biological variability. That said, we do observe a greater tendency for causal models to infer specific transitions at particular times (e.g., G1/S near 6h, as noted by Reviewer 482n), an effect that is largely mitigated by the transformer. We now address this more explicitly in the manuscript:
>
> *As observed in Supp. Figure 6  the timing of these checkpoints is highly variable. Sequence models effectively capture this variability, with transformer predictions being more accurate and therefore better aligned with the ground-truth distribution. In contrast, other methods tend to underestimate such variability.*
>
> **Highlighting the application areas would be useful**
>
> This is a good suggestion. We identify at least three major application areas for these models:
>
> 1) Fundamental interest in cell cycle analysis: Understanding cell cycle staging is crucial for studying growth, differentiation, metabolism, and circadian rhythms in biological systems.
> 2) Cancer cell line comparisons: Identifying intrinsic differences in proliferation potential across cancer cell lines.
> 3) Drug response prediction: Expanding training on multi-drug datasets could enhance predictions and enable in-silico drug screening.
>
>
> We have added a few sentences about the possible applications in the discussion of the  manuscript:
>
>
> *However, our findings demonstrate that sequence models can be effective predictors of cellular dynamics in more controlled settings,  providing a powerful tool for studying proliferation, cancer dynamics, and addressing cell cycle-related confounding factors. With larger datasets, models could improve drug effect predictions, paving the way for in-silico screening and broader biomedical applications.*

---

### Author Rebuttal · Authors · 2025-03-08

**Rebuttal:**

We appreciate the reviewers overall positive assessment, valuable suggestions and detailed comments!

In the revised manuscript (attached), we have made the following updates:

- Biological variability vs. memorization: Added Supp. Fig. 6, illustrating correlations between measured and predicted phase transition timings.
- Expanded discussion: Included more details on applicability, cell type diversity, and potential architecture extensions.
- Figure update: Fig. 1c now includes nuclear signal/area over time.
- Supplementary clarification: Added a section detailing checkpoint calculations.
- Fixed all typos and grammatical mistakes.

All individual reviewer comments are addressed in the sections below.

**Supporting Material:**

/attachment/17ff670a21d2fc2aa8b83ea0ecccdb6c7db468db.pdf

---

### Comment · Area_Chair_eC4i · 2025-03-12
**Discussion period**

Dear reviewers:

This paper was broadly positively received by a consensus. If there are no changes in opinion, please take a few moments to consider the paper's rebuttal arguments, clarify any remaining questions in the discussion period, and enter in your final evaluations. Thanks again for your time and expertise!

---

### Meta-Review · Area_Chair_eC4i · 2025-03-20

**Recommendation:** Accept (Oral)
**Confidence:** 5

**Metareview:**

This work attempts to use sequence models to predict fluorescent biomarkers using only simple light/brightfield microscopy. The paper was broadly enthusiastically received by the reviewers and should be presented at the conference.